# Effects of a Food Supplement Containing Hydrolyzed Collagen on Pain Perception, Joint Range, and Quality of Life in People with Chronic Knee Pain

**DOI:** 10.3390/diseases13070189

**Published:** 2025-06-20

**Authors:** Juan Carlos Salinas-Camargo, Cristian Marín-Pagán, Rosario Victoria Álvarez-Gil, Francisco Javier Martínez-Noguera, María Cabrera-Cabrera, Josep Manuel Llabrés-Laguarda, María Isabel Vasallo-Morillas, Pedro E. Alcaraz

**Affiliations:** 1Research Center for High Performance Sport, Catholic University of Murcia (UCAM), 30107 Murcia, Spain; jcsalinas2@ucam.edu (J.C.S.-C.); rvalvarez@ucam.edu (R.V.Á.-G.); palcaraz@ucam.edu (P.E.A.); 2Plantas Medicinales y Complementos Alimenticios, S.A. (PLAMECA), Av. Prat de la Riba, S/N, 08780 Pallejà, Spain; desarrolloproductos@plameca.com (M.C.-C.); jmllabres@plameca.com (J.M.L.-L.); 3San Antonio Technologies S.L., Avda. de los Jerónimos 135, 30107 Murcia, Spain; mivasallo@sat.ucam.edu; 4Faculty of Pharmacy, Catholic University of Murcia (UCAM), 30107 Murcia, Spain

**Keywords:** arthritis, joint pain, supplements, quality of life, range of motion

## Abstract

Background: Joint pain can impair joint function and limit a person’s ability to perform basic tasks and quality of life. The most used treatment is the pharmacological one. An alternative is the use of collagen-based food supplements. However, it remains a challenge to continue to develop new formulations to improve their efficacy. Objectives: The aim was to evaluate the long-term effectiveness of a food supplement based on hydrolyzed collagen alongside other active ingredients on knee joint pain, range of motion, and quality of life questionnaires in a moderately active population. Methods: A randomized, double-blind, and controlled study with two arms was completed with 80 participants, who took the prescribed supplementation (Curarti^®^ Selectium) or placebo for 40 days. Results: The supplement group showed a reduction of pain felt just after waking up in the morning. A statistically significant reduction in felt pain 3 h after exercise was observed at week 6 for the new product formula group compared to the placebo group. The symptoms associated with knee problems (KOOS) showed significant differences between groups. In functional capacity (WOMAC), it was found that the improvement was greater in the group treated with investigational formula (IF) than in the placebo group. The 36-item short form health survey (SF-36) about quality of life showed that the individuals who took IF improved with respect to those who took a placebo. Conclusions: The intake of Curarti^®^ Selectium for 40 days is effective in reducing joint pain at rest and after physical exercise, as well as maintaining the perception of quality of life, while allowing the physical functionality of the joint.

## 1. Introduction

Joint pain is defined as discomfort or tenderness in one or more joints. Acute joint pain is any joint pain that is expected to resolve within 6–8 weeks; chronic pain persists beyond this window [1].

Joint pain is common and usually felt in the hands, feet, hips, knees, or spine. The most common cause of chronic knee pain is degenerative osteoarthritis (OA) [2]. Other etiologies of knee pain include rheumatoid arthritis, crystal and spondylo-arthropathies, post-traumatic pain, and persistent postsurgical pain [3]. The pathogenesis of OA results from an interplay of biomechanical, inflammatory, and metabolic factors that contribute to cartilage damage, synovitis, and various subchondral bone abnormalities [4,5,6,7]. The underlying pain of this discomfort may be constant or intermittent, and the joint may sometimes feel stiff, sore, or irritated. Joint pain can impair joint function and limit a person’s ability to perform basic tasks. In severe cases, joint pain can also affect quality of life. Joint pain is also a common phenomenon in people who use or overuse a muscle repeatedly, suffer from depression, anxiety, or stress, or are overweight, among others [8]. Age also influences joint stiffness and pain. Hence, after years of wear and tear, middle-aged and older adults are typically more prone to experience symptoms of joint-related pain or discomfort.

Treatment for joint pain should not only focus on reducing pain, but maintaining mobility and reducing disability as well [8]. The most commonly used pharmacological treatment are: acetaminophen (higher dosage can produce nephropathy and increased risk of gastrointestinal (GI) bleeding, in addition, it is contraindicated in patients with liver disease or alcohol use), nonsteroidal anti-inflammatory drugs (NSAIDs) and cyclo-oxygenase-2 (COX-2) inhibitors (they have some contraindications like as peptic ulcer disease, GI bleed, renal disease, liver disease, and sensitivity to aspirin) [9], the opioids (they have some side effects like as GI and sedation) [10]. However, this type of treatment may improve symptoms, but not prevent the progression of the disease that causes the pain [11].

A more recent, vastly investigated alternative to pharmacological treatments is the use of collagen-based food supplements. The hydrolyzed collagen is a form of collagen that is processed intensively to break up the large collagen molecules into smaller fragments to increase absorption. Hydrolyzed collagen and gelatin may be the same in terms of amino acid composition, but they possess different chemical properties. Collagen is a native protein molecule with a molecular weight of ~300 kDa [12]. To produce hydrolyzed collagen, native collagen undergoes denaturation followed by a hydrolysis process, resulting in very low molecular mass (3–6 kDa) collagen peptides [12].

Different processing and post-processing methods to make collagen hydrolyzate can yield vastly different products, creating different collagen peptide sequences and molecular weights. These differences can potentially impact biological function in terms of regulating joint inflammation and the effect on subchondral bone. Furthermore, lower molecular weight collagen peptides may be more easily absorbed in the small intestine, theoretically increasing the likelihood of being delivered to other areas in the body, including joints. The resistance of collagen peptides to hydrolysis and digestion is primarily based on amino acid composition. In that regard, peptides with the amino acid proline or hydroxyproline are not readily hydrolyzed or digested by the gastrointestinal system, which may allow them to be absorbed in the small intestine [13,14].

The food supplements containing collagen have been recognized as safe by the Food and Drug Administration (FDA) and the European Food Safety Authority (EFSA) [15,16]. In addition, a 2017 meta-analysis of 69 clinical trials concluded that it is safe for use as a dietary supplement in humans to treat joint pain [17].

Of the 28 types of collagens, type II is one of the most abundant, accounting for 50% of the cartilage protein in joints. Researchers observed that when hydrolyzed collagen is administered orally, it is absorbed in the intestine and stimulates chondrocytes to produce type II collagen [18,19]. Some researchers even suggest that type II collagen has the potential to repair or regenerate damaged collagen [20]. In a 2006 study, 100 volunteers who experienced hip, shoulder, or knee pain associated with physical activity were given 10 g of hydrolyzed collagen daily for 12 weeks. As a result, 78% of participants reported reduced pain in the affected joint [18]. Another study [20] in patients with knee osteoarthritis found that those who ingested 10 g of hydrolyzed collagen per day had significantly less pain, as measured by a visual analogue pain scale (VAS), compared to placebo. In addition, there was an improvement in their score on the Western Ontario and McMaster Universities Arthritis Index (WOMAC). Pain improvement during physical activity is a measure used to assess pain reduction in the knee joint. This parameter was improved in participants in a randomized, double-blind clinical trial who had taken a formula based on hydrolyzed collagen compared to participants taking a placebo [21].

Knee range of motion (ROM) is essential to daily function for normal active individuals. Interventions aimed at improving ROM have been shown to relieve joint stiffness, increase joint mobility, and maintain joint function [21].

Various formulations of collagen-based products are currently available. However, it remains a challenge to continue to develop new formulations to check if there are improvements in efficacy with new products. In this sense, the primary aim of the present study was to evaluate the long-term effectiveness of a food supplement based on hydrolyzed collagen alongside other active ingredients on knee joint pain, knee range of motion, and quality of life questionnaires in a moderately active population. Therefore, we hypothesize that long-term intake of a collagen supplement together in combination with active ingredients, may improve knee pain and other quality of life parameters in moderately active subjects.

## 2. Methodology

### 2.1. Study Design

This was a single-center, randomized, double-blind, and controlled study with two arms. The study protocol was approved by the Ethics Committee of Universidad Católica San Antonio de Murcia (code CE012308 of 9 January 2023, Murcia, Spain). The protocol adhered to the ethical guidelines outlined in the World Medical Association’s Declaration of Helsinki 2013 [22]. In addition, the study was registered in https://clinicaltrials.gov/study/NCT05917925 (accessed on 7 June 2023) under the number NCT05917925. The study was conducted from 28 March 2023 to 2 February 2024. The required sample size was calculated using the G*POWER software version 3.1.9.7 (University of Düsseldorf, Düsseldorf, Germany). The analysis was based on an F-test for repeated measures ANOVA with a within-between interaction design, conducted a priori. The parameters set were: effect size f = 0.25, significance level (α) = 0.05, and statistical power (1 − β) = 0.95. Based on these inputs, the software estimated a total sample size of 54 participants as adequate. The sample size consisted of 81 participants (It was adequate to be representative of the population), of which one volunteer didn’t complete the study, so her data was not taken in the analysis, who took the prescribed supplementation or placebo for 40 days like similar studies [23] (Figure 1).

#### 2.1.1. Inclusion and Exclusion Criteria

Individuals included in this study met all the inclusion criteria listed below, and did not meet any of the exclusion criteria also listed below: (I) Adults from 18 until 60 years old (II) individuals who engaged in regular physical exercise and met the Word Health Organization (WHO) definition of being moderately active (Engage in moderate-intensity aerobic physical activity for at least 150 to 300 min per week or vigorous-intensity aerobic physical activity for at least 75 to 150 min per week, or an equivalent combination of moderate-intensity and vigorous-intensity activities more than 150 min throughout the week) [24], (III) persistent exercise-associated knee pain lasting at least 2 months prior to study entry, (IV) knee pain score on the visual analogue pain scale of at least 30 mm, taking into account that the scale has a maximum range of 100 mm. Exclusion criteria were: (I) people with severe disease or allergy to any of the product ingredients, (II), subjects with chronic inflammatory diseases affecting the musculoskeletal system, (III) BMI ≥ 30 kg/m^2^, (IV) treatment with narcotics, steroidal anti-inflammatory drugs or immunosuppressants, (V) symptoms of extreme pain requiring high doses of analgesic therapy for a period longer than 2 weeks or intra-articular injection treatment, (VI) pregnant or lactating women.

#### 2.1.2. Randomization

Once it was determined that the volunteers met all the inclusion criteria and none of the exclusion criteria, the informed consent was signed, and from that moment the volunteer would be considered included in the study.

During the first visit (day 1), the participant was randomly assigned to one of the two study groups following the randomization code list generated by a specific software (Epidat 3.1), which was developed by a project monitor who did not participate in the experimental stage and data analysis of the study. This list used a stratified randomization process by age (ratio 18–30 years: 31–49 years: 50–60 years = 40:40:20). Likewise, each stratum divided the participants into 2 groups, according to the assigned product (placebo and IF). Forty-one subjects participated in the placebo group and thirty-nine in the IF group.

#### 2.1.3. Research Product

The investigational products involved in this study were a hydrolyzed collagen-based food supplement (Curarti^®^ Selectium, Plantas Medicinales y Complementos Alimenticios S.A, Barcelona, Spain) and a placebo.

The ingredients of the food supplement were the following: hydrolyzed collagen, magnesium carbonate, vitamin C (L-ascorbic acid), Curcuma longa root extract (standardized in 95% curcuminoids), hyaluronic acid (as sodium salt), resveratrol, and anti-caking agent (silicon dioxide).

The placebo was composed of the following ingredients: Maltodextrin, soy lecithin, rice flour, powdered sugar, and food coloring (corn flour, salt, tartrazine (E-102), and allura red AC (E-129).

The IF and placebo were identical in appearance, smell, color, and weight. This ensured that participants did not distinguish between the two. Each participant in the study randomly received a 500 g container of which he/she should ingest 10 g of product per day (in the morning with breakfast) for 6 weeks based on similar trails [20,25,26], following the preparation instructions indicated by the manufacturer (10 g dissolved in a glass of water or fruit juice).

#### 2.1.4. Assessments

##### Joint Pain Intensity

It was measured using a 100 mm long horizontal visual analog scale (VAS) on which the participant had to mark the pain he/she felt with 0 being “none” and 100 being “the most intense pain he/she could imagine”. Once the pain level was marked on the horizontal line, the length was measured from 0 in millimeters with a millimeter ruler like similar studies [20,26]. The results were evaluated considering three levels of pain according to the marked values: <40 mm (mild or mild-moderate pain); 40–60 mm (moderate to moderate-severe), and >60 mm (considered severe to unbearable).

The intensity of joint pain was measured: (I) Daily: the participant had to mark on the scale the maximum pain felt the previous day, first thing in the morning (30 min after waking up), (II) After physical exercise: During the hour after the physical activity performed, the participant had to mark the maximum pain felt during physical exercise. Then, the participant should mark the pain felt 3 h after the physical activity. (III) During visits to the research center, the investigator gave the participant a VAS to mark the average pain felt during the last week.

##### Knee Range of Motion (ROM) by Goniometry

To assess ROM, the participant was placed in a prone position on a stretcher. Subsequently, a K-move electrogoniometer (Kinvent, Montpellier, France) was placed on the leg being assessed (affected leg), and the participant was asked to perform voluntary flexion up to the maximum possible point or until pain was felt. After such flexion, the ROM angle was recorded and the measurement was repeated a total of 3 times, separated by 45 s. The final value to be considered was the mean of the 3 ROMs obtained. Calibration of the apparatus was performed at each measurement by marking “zero” at the beginning of the recording and did not require additional calibration.

WOMAC scale to measure pain, functional capacity, and the impact on quality of life.

The WOMAC (Western Ontario and McMaster Universities Osteoarthritis Index) questionnaire in its Spanish version [27] was used to evaluate the changes perceived by the participants in relation to the degree of pain, stiffness, and functionality of the knee. The reduced version of the WOMAC questionnaire with 11 items was chosen because it had improved properties over the original 24-item WOMAC questionnaire [28] and also has the advantage of optimizing the time of the study visits, as it can be answered more quickly.

The reduced WOMAC questionnaire consists of 11 questions: three assess pain, two assess stiffness, and six assess functionality. Responses to the questions are scored on a scale of 0 to 10, with 0 being no pain, stiffness, and difficulty, and 10 being maximum pain, stiffness, and difficulty.

##### Quality of Life Questionnaire SF-36

The SF-36 health questionnaire [29] is composed of 36 items that aim to collect all relevant aspects to characterize the health of an individual. These questions cover at least 8 aspects or dimensions: Physical Function, Physical Role; Bodily Pain; General Health; Vitality; Social Function; Emotional Role, and Mental Health. For each of these dimensions, easily interpretable score scales can be calculated, all of them characterized by being ordered in such a way that the higher the value obtained, the better the state of health. We used the Spanish version in this study [30].

##### KOOS Questionnaire for Knee Evaluation

The Knee Injury and Osteoarthritis Outcome Score (KOOS) [31] was developed as an instrument to assess the patient’s opinion on their knee and its associated problems. The KOOS questionnaire in its Spanish version [32] has been widely used for research in clinical trials, large-scale databases, and registries. It is also useful in daily clinical practice to monitor groups and individuals over time. KOOS assesses 5 subscales: pain, other symptoms, activities of daily living, function, and sports/recreational activities, and quality of life.

The resulting scores are transformed into a scale from 0 to 100, where 0 represents extreme knee problems and 100 represents no knee problems.

### 2.2. Statistical Methods

The statistical analysis was conducted using IBM Social Sciences software (SPSS, v.21.0, Chicago, IL, USA). Mean ± SD was used to present the data. The Levene and Shapiro–Wilk tests were employed to check the homogeneity and normality of the data, respectively. A Two-way repeated-measures ANOVA was used to analyze the results of knee ROM and WOMAC, SF-36, and KOOS questionnaires, with factors including time [PRE (day 1) vs. POST (day 40)] and condition (placebo or supplement group). In the case of significance in ANOVA models, Tukey’s post hoc analysis was carried out. Partial eta squared (ηp2) was also calculated as an effect size for time, condition, and time × condition interaction of all variables in the ANOVA analysis. The thresholds for partial eta squared were applied as follows: <0.01, irrelevant; ≥0.01, small; ≥0.059, moderate; ≥0.138, large [33]. The Significance level was set at *p* ≤ 0.05. Pearson’s correlation (r) was used to evaluate the correlations between the parameters.

## 3. Results

### 3.1. Disposition of the Subjects

The following diagram (Figure 2) shows subject flow from being recuiting until analyzed.

### 3.2. Baseline Characteristics of the Population

The individuals included in this study had a mean age of 35.1 ± 12.1 years. A total of 49.4% were men and 50.6% were women. The mean body weight of these individuals was 72.2 ± 13.5 kg and height 172 ± 10.7 cm, giving a mean BMI of 24.3 ± 2.7. Therefore, the BMI of the population studied was in the normal weight range.

After being randomized to the two treatment groups, there were no differences in age, weight, height, and BMI between the two groups. The distribution of males and females in each group was also the same.

Therefore, it can be concluded that the sample of individuals studied was homogeneous in terms of their demographic characteristics (Table 1).

### 3.3. Measurement of the VAS Pain Scale 30 min After Waking Up

It was observed that the evolution of pain at 40 days showed a significant reduction (*p* < 0.05) in the group taking IF, while the reduction in the group taking the placebo was not significant. Furthermore, the difference between the two groups was almost significant (*p* = 0.082), indicating that treatment with IF tends to significantly reduce the pain felt upon waking up each morning, after 40 days of treatment (Table 2).

Furthermore, from week 4 onwards, weekly differences very close to statistical significance were observed between the two treatment groups; the statistical significance was obtained at week 6 (result with *) with a clear reduction in pain in the group that received IF with respect to the group placebo (Table 3). These results indicate that the administration of the collagen-based food supplement has an effect on the reduction of pain felt just after waking up in the morning.

### 3.4. Measurement of the VAS Pain Scale 3 h After the Physical Activity Performed

Table 4 shows the mean results of perceived pain 3 h after physical activity.

The mean weekly value of pain felt after 3 h of physical activity was calculated and compared with baseline (Table 5). Each week, the difference from baseline was statistically significant (*p* < 0.05) in the two treatment groups. The comparisons made between the groups each week showed no significant differences except for week 6, when, considering the time factor and the treatment factor, the IF group reduced pain significantly (*p* < 0.05) with respect to the placebo.

### 3.5. KOOS Questionnaire Results for Knee Evaluation

Table 6 shows the mean and standard deviation of the domains of the KOOS questionnaire: pain, activities of daily living, functionality, sports and recreational activities, quality of life, and symptoms measured at the study visits.

For pain, an increase in this variable was observed over time, which implies an improvement in pain in both groups (IF and placebo) (Table 6). Furthermore, when comparing the results of each visit with respect to the baseline (Table 6), significant differences were obtained (*p* < 0.05) for both the IF group and the placebo group. However, when comparing the groups, they did not differ significantly.

For daily activities, it is observed that in both treatment groups, the results increase as the study progresses. However, when comparing the results between each visit and the baseline visit (Table 6), significant differences are obtained (*p* < 0.05), but when comparing the two treatment groups, no differences are found between them.

For functionality, recreational activities, and sports, the results increase in both treatment groups (Table 6). However, when comparing the results between each visit and the baseline visit, significant differences are obtained (*p* < 0.05), but when comparing the two treatment groups, no differences are found between them (Table 6).

The results obtained (Table 6) for each of the dimensions measured in the KOOS questionnaire have shown a significant reduction in knee problems over time, in both treatment groups. The reduction in knee problems was greater in all dimensions in the group treated with IF, although only the dimension related to symptoms associated with knee problems (KOOS symptoms) showed significant differences (*p* < 0.05) between the treatment groups.

### 3.6. WOMAC Osteoarthritis Scale

Differences were observed (Table 7) in the functional capacity results between visit 2 and the initial visit. There were significant differences (*p* < 0.05) as a function of time, and taking into account the time and treatment factor, showed an improvement in functional capacity in the group treated with IF with respect to the group treated with placebo.

The results showed a reduction in the score of each domain at visit 2 for the two treatment groups (Table 7). The reduction was significant (*p* < 0.05) in both groups, but it was greater for the IF-treated group in all the domains analyzed.

In the functional capacity domain, it was found that the improvement was significantly (*p* < 0.05) greater in the group treated with IF with respect to the placebo group.

### 3.7. SF-36 Health Questionnaire

The general health of the participants who took the placebo product worsened statistically significantly (*p* < 0.05) during the study (Table 8). This was not observed in the participants treated with the IF product. Furthermore, when comparing the general health results obtained for both groups, a significant difference (*p* < 0.05) was obtained between the two treatments.

The following table shows that the quality of life of the individuals who took the collagen-based product improved significantly (*p* < 0.05) with respect to those who took a placebo, taking into account the time factor and the treatment factor.

### 3.8. Knee Range of Motion (ROM) by Goniometry

The participant performs a voluntary flexion of the leg to the maximum possible point or until pain is felt. The higher the ROM angle value, the greater the flexion. Therefore, an increase in the ROM angle indicates improvement (Table 9).

On day 40 of the study, there was a significant (*p* < 0.05) decrease in ROM angle in the placebo-treated group, thereby indicating a decrease in knee flexion amplitude and thus a worsening of the participants’ condition. IF-treated individuals slightly increased the ROM angle, but this increase was not significant. However, when the two treatment groups were compared, a trend towards statistical significance (*p* = 0.072) of the difference was obtained. It could be concluded that the intake of the collagen-based food supplement prevents the progression of the worsening of the condition suffered by the affected knee.

## 4. Discussion

The main objective of this study was to evaluate the effect of chronic supplementation with a food supplement based on hydrolyzed collagen and other active ingredients on knee joint pain in a moderately active population. The main findings of this study were an improvement in the WOMAC, KOOS, SF-36, and EVA questionnaires after the end of the IF supplementation period. This is in line with the hypothesis generated, where we established that chronic IF intake improved quality of life and pain parameters in moderately active subjects.

In the present study, analysis of the primary outcome measure, which was the proportion of subjects who experienced a reduction in pain as defined by a >30 mm decrease in VAS, (Table 5) showed a significant decrease 30 min after standing up (*p* = 0.022) and 3 h after exercise (*p* = 0.019) in the IF-supplemented group (Table 7). Our results are in line with those found in previous studies (controlled clinical trials), which have shown that patients with different osteoarthritic conditions can improve the subjective symptoms of arthritic conditions following the intake of 10 g daily of IF [34,35,36,37,38].

In addition, we also found improvements in the WOMAC test after chronic IF supplementation. In line with these results, a similar study [39] also found that hydrolyzed collagen supplementation improved WOMAC score in thirty subjects diagnosed with knee osteoarthritis. However, the main difference with this study is that we excluded patients with diagnosed osteoarthritis since our goal was to evaluate the effect of IF supplementation in the general population. Furthermore, when comparing our results with other studies, it’s important to highlight the fact that they evaluated the effect of the intake of collagen on joint problems simultaneously with non-steroidal analgesics, analgesics, antipyretics, corticosteroids, and this may introduce bias in the conclusions obtained [18,19,20,40]. Therefore, based on our results in the VAS and WOMAC tests, chronic supplementation with the IF can improve perceived pain in the active population.

On the other hand, we also found improvements in the score of KOOS after chronic IF supplementation. Some other authors [26,41] also found similar results, particularly a study published in 2023 [42] saw a difference in the mentioned questionnaire in people who exercised 188 min per week. Physical exercise can reduce pain with collagen supplementation because it can increase blood flow and the supply of amino acids and bioactive peptides to the connective tissue (low vascularity) during exercise. The pain-mitigating exercise-moderating effects of collagen supplementation may be attributed to increased blood flow and delivery of bioactive amino acids and peptides to connective tissue during exercise, which is otherwise poorly vascularized [43,44].

No improvement in knee ROM is seen after the supplementation period, but this is nevertheless a positive result as it maintains anatomical functionality and prevents deterioration, as opposed to the placebo, which shows some loss of functionality in ROM. The analysis shows a tendency to change between groups, so we can interpret that in longer periods of time or larger samples, clearer differences could be observed. This shows that the normal course of the ROM (Placebo) would worsen with time [18], so that the investigational product would have a positive effect even if there is no improvement. In previous studies, several researchers [18,34] also found similar results for functional assessments applying mobility and flexibility tests, but in this case, they also found no differences or trends of change between groups. This result is important as it may indicate that IF may help to slow the functional deterioration of the knee joint.

With respect to the subjective evaluation of the perceived quality of life by means of the SF-36 questionnaire, we can find that the participants supplemented with IF showed improvements in the questions referring to physical function, both when comparing the effect of the treatment over time and in the comparison with the placebo group. In the questions corresponding to other aspects of quality of life recorded by the questionnaire, no notable changes were found. These findings are in the same direction as those found by previous research [29,45] that also report improvements in perceived physical function. Therefore, it can be stated that treatment with the IF supplement for 6 weeks improves the quality of life related to physical activity.

With the results obtained, we can consider that the IF used in this study can contribute to reducing knee joint pain in moderately active and otherwise healthy people with chronic pain, improving some aspects of their quality of life and functionality. These results contribute to corroborating other research with collagen-based products.

This study has several strengths, including the use of a product with a high total collagen content and a rigorous, double-blind, randomized, placebo-controlled study design. The main statistical methods and analyses were robust, which is further confirmed by a sensitivity analysis. Additionally, the study population included healthy people with general joint pain, stiffness, and lack of mobility, rather than patients diagnosed with OA alone.

### Limitations of the Study and Future Directions

Some important limitations of this study were the duration of 40 days and the inclusion criteria of mild-moderate pain (40 mm out of 100 on the VAS scale). Additional studies are needed to further evaluate the long-term effects of the test product consumption on biochemical markers of inflammation, joint collagen formation and destruction, to have a more robust understanding of the effects of collagen supplementation on the general discomfort of the joints.

## 5. Conclusions

The intake of Curarti^®^ Selectium for 40 days (6 weeks approximately) is effective in reducing joint pain at rest and after physical exercise, as well as maintaining the perception of quality of life, while allowing the physical functionality of the joint. It prevents a worsening of the condition in healthy people with chronic knee joint pain without the need for medical treatments.

## Figures and Tables

**Figure 1 diseases-13-00189-f001:**
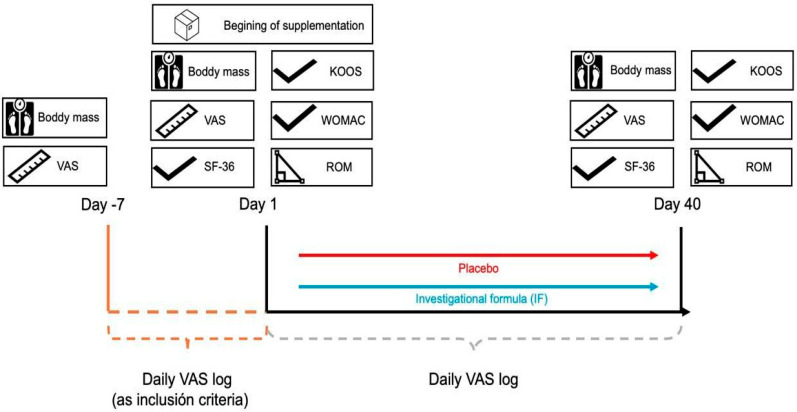
Study timeline. VAS: Visual analogue pain scale; SF-36: Quality of life questionnaire; KOOS: The knee injury and osteoarthritis outcome score; WOMAC: Western Ontario and McMaster Universities Osteoarthritis Index; ROM: Knee range of motion.

**Figure 2 diseases-13-00189-f002:**
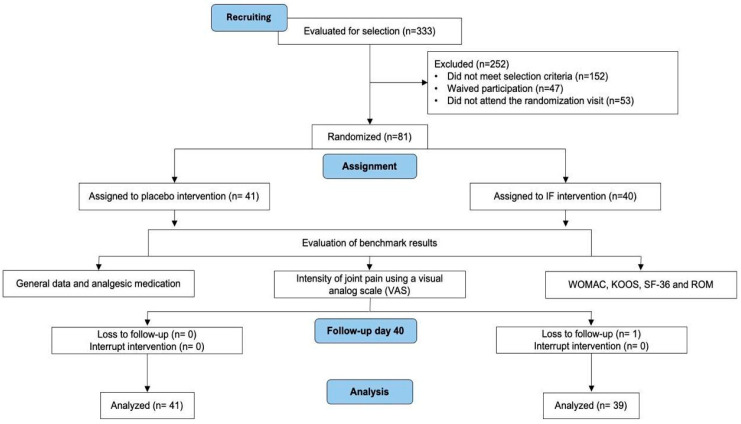
Flow chart of study participants. VAS: Visual analog scale; WOMAC: Western Ontario and McMaster Universities Osteoarthritis Index; KOOS: The Knee injury and Osteoarthritis Outcome Score; SF-36: The SF-36 health questionnaire; and ROM: Knee range of motion.

**Table 1 diseases-13-00189-t001:** Demographic characteristics of the study population.

	IF	Placebo	Overall
	(N = 39)	(N = 41)	(N = 80)
Age (years)			
Mean (SD)	35.6 (12.3)	34.7 (12.1)	35.1 (12.1)
Median [Min, Max]	35.0 [18.0, 58.0]	31.0 [18.0, 60.0]	32.0 [18.0, 60.0]
Gender			
Male	19 (47.5%)	21 (51.2%)	40 (49.4%)
Female	21 (52.5%)	20 (48.8%)	41 (50.6%)
Height (cm)			
Mean (SD)	170 (9.71)	174 (11.2)	172 (10.7)
Median [Min, Max]	170 [154, 196]	174 [153, 198]	170 [153, 198]
Weight (Kg)			
Mean (SD)	71.2 (12.5)	73.2 (14.6)	72.2 (13.5)
Median [Min, Max]	70.4 [49.0, 109]	74.8 [49.8, 100]	71.3 [49.0, 109]
BMI			
Mean (SD)	24.6 (2.48)	24.0 (2.90)	24.3 (2.70)
Median [Min, Max]	24.3 [20.4, 29.2]	24.3 [17.7, 29.2]	24.3 [17.7, 29.2]

IF: Investigational formula; N: population; SD: standard deviation; cm: centimeters; Min: minimum; Max: maximum; BMI: Body mass index.

**Table 2 diseases-13-00189-t002:** Evolution of pain felt 30 min after standing up, measured with the pain VAS at 40 days with respect to baseline pain.

Day	Group	Mean	95% CI_Low	95% CI_High	*p* Value *
40	IF	−1.089	−1.776	−0.402	0.002
	Placebo	−0.203	−0.932	0.526	0.580
	Differences	0.886	−0.116	1.887	0.082
	g of Hedges	−0.42	−0.91	0.06	small

IF: Investigational formula; CI: Confidence interval; *: *p* < 0.05 statistically significant.

**Table 3 diseases-13-00189-t003:** Evolution of pain felt 30 min after standing up measured with the VAS of pain on a weekly basis with respect to pain at week 1. The difference is the difference between the increases obtained for the IF group and the placebo group each week. Statistical significance is obtained between groups at week 6 (result with *).

	Group	Mean	95% CI_Low	95% CI_High	*p* Value *
Week 2	IF	−0.482	−0.805	−0.160	0.004
	Placebo	−0.452	−0.766	−0.137	0.005
	Differences	0.031	−0.419	0.481	0.892
Week 3	IF	−0.803	−1.201	−0.405	0.000
	Placebo	−0.606	−0.994	−0.218	0.003
	Differences	0.197	−0.359	0.753	0.482
Week 4	IF	−1.120	−1.572	−0.668	0.000
	Placebo	−0.556	−0.997	−0.115	0.014
	Differences	0.564	−0.068	1.196	0.080
Week 5	IF	−1.239	−1.702	−0.775	0.000
	Placebo	−0.639	−1.091	−0.186	0.006
	Differences	0.600	−0.048	1.248	0.069
Week 6	IF	−1.404	−1.878	−0.931	0.000
	Placebo	−0.623	−1.091	−0.155	0.010
	Differences	0.782	0.116	1.447	0.022 *

IF: Investigational formula; CI: Confidence interval; *: *p* < 0.05 statistically significant.

**Table 4 diseases-13-00189-t004:** Mean and standard deviation of pain felt 3 h after physical activity measured with the pain VAS on days 1 and 40 of the study.

Product	Measure	Mean	SD
IF	Day 1	4.7	2.0
	Day 40	2.6	1.9
Placebo	Day 1	4.4	1.9
	Day 40	3.4	2.6

IF: Investigational formula; SD: standard deviation.

**Table 5 diseases-13-00189-t005:** Increase in pain felt 3 h after physical activity measured with the VAS of pain on a weekly basis with respect to pain at week 1. The difference is the difference between the increases obtained for the IF group and the placebo group each week.

	Group	Mean	95% CI_Low	95% CI_High	*p* Value *
Week 2	IF	−0.342	−0.728	0.043	0.081
	Placebo	−0.519	−0.895	−0.143	0.007
	Differences	−0.177	−0.715	0.361	0.514
Week 3	IF	−0.934	−1.398	−0.469	0.000
	Placebo	−0.641	−1.099	−0.182	0.007
	Differences	0.293	−0.359	0.945	0.374
Week 4	IF	−1.039	−1.576	−0.502	0.000
	Placebo	−0.744	−1.281	−0.207	0.007
	Differences	0.295	−0.465	1.055	0.442
Week 5	IF	−1.393	−1.923	−0.863	0.000
	Placebo	−1.015	−1.538	−0.492	0.000
	Differences	0.377	−0.367	1.122	0.316
Week 6	IF	−1.827	−2.393	−1.262	0.000
	Placebo	−0.865	−1.431	−0.299	0.003
	Differences	0.962	0.162	1.763	0.019 *

IF: Investigational formula; CI: Confidence interval; *: *p* < 0.05 statistically significant.

**Table 6 diseases-13-00189-t006:** KOOS questionnaire results.

Domains	Product	Measure	Mean (SD)	95% CI_Low	95% CI_High	*p* Value *
Pain	IF	Day 1	65.2 (12.3)	−0.497	7.387	0.086
		Day 40	68.7 (15.1)			
	Placebo	Day 1	66.0 (16.5)	1.614	9.401	0.006
		Day 40	71.5 (16.1)			
	Differences		2.062	−3.479	7.603	0.461
	g of Hedges		−0.16	−0.6	0.28	negligible
Daily activities	IF	Day 1	67.3 (16.9)	2.933	11.162	0.001
		Day 40	74.3 (16.0)			
	Placebo	Day 1	73.1 (14.4)	2.031	10.159	0.004
		Day 40	79.2 (14.0)			
	Differences		−0.952	−6.735	4.830	0.744
	g of Hedges		0.07	−0.37	0.51	negligible
Functionality, sports and recreational activities	IF	Day 1	43.9 (19.4)	4.963	15.537	0.000
		Day 40	54.1 (20.0)			
	Placebo	Day 1	47.2 (20.6)	2.705	13.149	0.003
		Day 40	55.1 (21.8)			
	Differences		−2.323	−9.755	5.108	0.536
	g of Hedges		0.14	−0.3	0.58	negligible
Quality of life	IF	Day 1	48.6 (16.4)	7.808	17.202	0.000
		Day 40	61.1 (16.5)			
	Placebo	Day 1	52.5 (17.6)	2.353	11.631	0.004
		Day 40	59.5 (18.6)			
	Differences		−5.513	−12.114	1.089	0.100
	g of Hedges		0.37	−0.08	0.81	small
Symptoms	IF	Day 1	39.0 (9.1)	5.482	12.993	0.000
		Day 40	48.2 (12.7)			
	Placebo	Day 1	42.0 (11.2)	−0.217	7.202	0.065
		Day 40	45.5 (11.6)			
	Differences		−5.745	−11.024	−0.466	0.033 *
	g of Hedges		0.48	0.03	0.92	small

IF: Investigational formula; SD: Standard deviation; CI: Confidence interval; *: *p* < 0.05 statistically significant.

**Table 7 diseases-13-00189-t007:** WOMAC questionnaire results.

Domains	Product	Measure	Mean (SD)	95% CI_Low	95% CI_High	*p* Value *
Pain	IF	Day 1	4.3 (2.3)	−1.893	−0.707	0
		Day 40	3.0 (1.7)			
	Placebo	Day 1	3.6 (2.2)	−1.122	0.049	0.072
		Day 40	3.1 (2.0)			
	Differences		0.763	−0.070	1.597	0.1475
	g of Hedges		−0.4	−0.84	0.04	small
Stiffness	IF	Day 1	3.2 (1.5)	−1.103	−0.097	0.020
		Day 40	2.6 (1.8)			
	Placebo	Day 1	2.6 (1.4)	−0.765	0.228	0.286
		Day 40	2.4 (1.8)			
	Differences		0.332	−0.375	1.038	0.353
	g of Hedges		−0.21	−0.64	0.23	small
Functional capacity	IF	Day 1	6.4 (3.2)	−2.703	−1.047	0.000
		Day 40	4.5 (2.7)			
	Placebo	Day 1	4.9 (3.3)	−1.525	0.111	0.089
		Day 40	4.2 (2.9)			
	Differences		1.168	0.004	2.332	0.049 *
	g of Hedges		−0.44	−0.88	0	small

IF: Investigational formula; SD: Standard deviation; CI: Confidence interval; *: *p* < 0.05 statistically significant.

**Table 8 diseases-13-00189-t008:** SF-36 questionnaire results.

Domains	Product	Measure	Mean (SD)	95% CI_Low	95% CI_High	*p* Value *
General health	IF	Day 1	36.5 (9.0)	−2.991	2.791	0.945
		Day 40	36.4 (7.0)			
	Placebo	Day 1	36.5 (8.9)	−7.235	−1.380	0.004
		Day 40	33.6 (8.4)			
	Differences		−4.208	−8.322	−0.093	0.045 *
	g of Hedges		−0.29	−1.01	0.44	small
Physical function	IF	Day 1	71.6 (17.2)	5.485	14.265	0.000
		Day 40	82.0 (15.3)			
	Placebo	Day 1	82.0 (17.6)	−2.384	6.287	0.373
		Day 40	83.8 (15.4)			
	Differences		−7.924	−14.094	−1.754	0.012 *
	g of Hedges		0.56	0.12	1.01	medium
Body pain	IF	Day 1	37.3 (13.0)	−11.847	−1.153	0.018
		Day 40	30.8 (15.1)			
	Placebo	Day 1	37.7 (14.7)	−11.038	−0.474	0.033
		Day 40	32.0 (16.2)			
	Differences		0.744	−6.772	8.260	0.844
	g of Hedges		−0.04	−0.48	0.39	negligible
Vitality	IF	Day 1	50.0 (8.9)	−5.920	0.920	0.150
		Day 40	48.0 (8.5)			
	Placebo	Day 1	45.9 (10.8)	−2.646	4.110	0.668
		Day 40	46.6 (7.9)			
	Differences		3.232	−1.575	8.038	0.185
	g of Hedges		−0.29	−0.74	0.15	small
Emotional role	IF	Day 1	83.0 (32.0)	−5.036	18.371	0.260
		Day 40	89.2 (29.0)			
	Placebo	Day 1	86.2 (29.8)	−9.931	13.190	0.780
		Day 40	87.8 (25.6)			
	Differences		−5.038	−21.489	11.412	0.544
	g of Hedges		0.13	−0.3	0.57	negligible
Physical role	IF	Day 1	59.7 (32.1)	−9.885	11.135	0.906
		Day 40	60.3 (35.8)			
	Placebo	Day 1	67.7 (36.8)	−3.064	17.698	0.165
		Day 40	75.0 (31.6)			
	Differences		6.692	−8.080	21.464	0.370
	g of Hedges		−0.2	−0.64	0.24	negligible
Social Function	IF	Day 1	47.0 (12.4)	−5.315	3.915	0.763
		Day 40	46.3 (14.2)			
	Placebo	Day 1	46.6 (11.9)	−2.424	6.692	0.354
		Day 40	48.8 (10.0)			
	Differences		2.834	−3.652	9.320	0.387
	g of Hedges		−0.19	−0.63	0.25	negligible
Mental health	IF	Day 1	52.2 (18.6)	−1.372	7.872	0.166
		Day 40	55.4 (12.1)			
	Placebo	Day 1	56.0 (11.6)	−4.517	4.614	0.983
		Day 40	56.0 (9.7)			
	Differences		−3.201	−9.698	3.296	0.330
	g of Hedges		0.22	−0.22	0.66	small

IF: Investigational formula; SD: Standard deviation; CI: Confidence interval; *: *p* < 0.05 statistically significant.

**Table 9 diseases-13-00189-t009:** Mean and standard deviation of knee range of motion (ROM) measured with a goniometer on days 1 and 40 of the study.

Product	Measure	Mean (SD)	95% CI_Low	95% CI_High	*p* Value *
IF	Day 1	119.4 (11.1)	−2.144	3.279	0.678
	Day 40	120.0 (11.2)			
Placebo	Day 1	124.1 (12.8)	−5.712	−0.220	0.035
	Day 40	122.6 (9.6)			
Differences		−3.533	−7.393	0.326	0.072
g of Hedges		−0.14	−0.58	0.3	negligible

IF: Investigational formula; SD: Standard deviation; CI: Confidence interval; *: *p* < 0.05 statistically significant.

## Data Availability

The data that support the findings of this study are available from the corresponding author upon reasonable request.

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
