# Peer review of "Effects of a Food Supplement Containing Hydrolyzed Collagen on Pain Perception, Joint Range, and Quality of Life in People with Chronic Knee Pain"

_diseases, 2025, doi:10.3390/diseases13070189_

Round 1
Reviewer 1 Report
Comments and Suggestions for Authors
I have the following suggestions for authors.
Comment-1-Title needs to be rewritten again to make it clearer. I suggest adding the term food supplement.
Comment-Line 23- “A statistically significant reduction in felt pain 3 hours 23 after exercise “- Write this sentence correctly.
Comment-3- Line 27-What is IF, S-36? Please write the full form first.
Comment-4-Line 29- What is Curarti® Selectium ? Should mention this first in the methodology portion of your abstract. Overall, it is hard to visualize the authors' work from the abstract. So, I suggest writing in a storytelling way to engage the readers.
Comment-5-In the first paragraph of introduction, authors should discussed in detail about the etiology of joint pain. Authors can make schematic diagram using bio render to explain all possible cause of joint pain.
Comment-6- Authors are just superficially claiming that recently available medicines are not efficient for joint pain. First authors should mention all major drugs used to treat joint pain and should validate scientifically why they have not been effective. Just saying is not sufficient. And then, they should write very briefly on available alternative approaches, followed by the utilization of collagen-based food supplements.
Comment-7- Line 51-57- Firstly, authors should discuss about what are collagen-based food supplement, what is their composition and how they are manufactured.
Comment-8- Why do authors have a total of 80 sample size of only? What sample size determination technique they used it should be shown here. Also, is an 80-sample size sufficient to reflect the clinical efficacy in humans?
Comment-9-Section 2.1.3.- On what basis do authors determine the dose of the placebo and sample? If they used the dose from previous studies, it should be cited. Through the methodology, they have not cited the protocols properly. Please cite every article if the authors have adopted the methodology.
Comment-10- Line 312- The way of citation is wrong. Please follow the journal guidelines and use only one citation style; i.e., [X].
Comment-11- In the discussion section, authors have only focused on their result. Authors should have in depth discussion to explore possible mechanism by which collagen-based product control the joint pain.
Comment-12-Authors are strongly suggested to have sub-sections of Limitations of study and future direction before the conclusion.
Comments on the Quality of English Language
Could be improved
Reviewer 2 Report
Comments and Suggestions for Authors
This is an interesting and well-developed study focusing on chronic knee pain in a moderately active population. Overall, the experimental design and analysis appear rigorous; however, several important issues still need addressing before publication.
Abstract:
- The use of acronyms is somewhat confusing. While KOOS and WOMAC are clear, “IF” is ambiguous. The authors should clearly define IF upon first usage.
Introduction:
- The manuscript title specifies “chronic” joint pain. Could the authors clarify the definitions and distinctions between chronic and acute joint pain, and explain explicitly why chronic joint pain was specifically chosen for the study?
- The authors stated: “However, this type of treatment may improve symptoms, but does not prevent the progression of the disease that causes the pain.” Does the tested supplement have the potential to prevent disease progression?
- Could the authors provide more details about the specific active ingredients included in the supplement and justify their necessity in this formulation?
Methodology:
- Important: How was the sample size determined?
- The inclusion/exclusion criteria lack age specifications. Are only adults included? Is there an upper age limit?
- It would be helpful for readability if the WHO definition of “moderately active” was concisely explained in the criteria part.
- Important: The authors should clearly explain how allocation concealment and blinding were conducted, as these are critical to the rigor of an RCT evaluating pain outcomes.
- Are the IF and placebo identical in appearance, smell, color, and weight, ensuring participants cannot distinguish between them?
- Regarding the timing of supplementation (“in the morning”), does consumption before or after a meal matter?
- Were all questionnaires provided and completed in English?
- Why was a follow-up period of 40 days specifically chosen?
Results:
- Acronym usage should consistently follow the standard format: the full name should be provided with its acronym at first mention, after which only acronyms are used. The manuscript also requires careful proofreading for grammatical errors and typos.
- Were all baseline characteristics and measures comparable between groups initially?
- In the provided tables, the term “difference” is ambiguous. Does it refer to between-group differences? Also, “IC95%” is less commonly used compared to the standard “95% CI.” Additionally, were statistical tests performed for Table 4?
- Were any side effects reported or observed for participants taking the supplement?
Discussion:
- The authors mentioned that the main distinction from the Kumar et al. study was the exclusion of osteoarthritis patients. Was this critical detail explicitly stated in the methodology section?
Round 2
Reviewer 1 Report
Comments and Suggestions for Authors
Dear authors, when responding to each comment, it is mandatory to indicate which page and line numbers represent any changes and additions. A reviewer will not be able to scan each line manually for every comment.
So, I suggest responding by showing the line numbers in the manuscript where changes and additions have been made.
Reviewer 2 Report
Comments and Suggestions for Authors
Thank you to the authors for their responses. While the major concerns have been addressed, I have two minor remaining points:
- If the questionnaires were administered in Spanish, please provide references for the translated versions. These instruments should undergo a process of cross-cultural adaptation and psychometric validation to ensure their reliability and validity in the target population.
- Regarding Question 12, please provide references or clinical evidence to support the rationale for selecting a 40-day follow-up period.
